# Proteomic Changes during the Dermal Toxicity Induced by *Nemopilema nomurai* Jellyfish Venom in HaCaT Human Keratinocyte

**DOI:** 10.3390/toxins13050311

**Published:** 2021-04-27

**Authors:** Indu Choudhary, Duhyeon Hwang, Jinho Chae, Wonduk Yoon, Changkeun Kang, Euikyung Kim

**Affiliations:** 1College of Veterinary Medicine, Gyeongsang National University, Jinju 52828, Korea; induchoudhary2u@gmail.com (I.C.); pooh9922@hanmail.net (D.H.); ckkang@gnu.ac.kr (C.K.); 2Institute of Animal Medicine, Gyeongsang National University, Jinju 52828, Korea; 3Marine Environmental Research and Information Laboratory, B1101, 17 Gosan-ro 148beon-gil, Gunpo-si 15850, Gyeonggi-do, Korea; jinhochae@gmail.com (J.C.); wondukyoon@humer.co.kr (W.Y.)

**Keywords:** *Nemopilema nomurai* jellyfish, 2-DE, MALDI-TOF/MS, HaCaT cell, dermal toxicity

## Abstract

Jellyfish venom is well known for its local skin toxicities and various lethal accidents. The main symptoms of local jellyfish envenomation include skin lesions, burning, prickling, stinging pain, red, brown, or purplish tracks on the skin, itching, and swelling, leading to dermonecrosis and scar formation. However, the molecular mechanism behind the action of jellyfish venom on human skin cells is rarely understood. In the present study, we have treated the human HaCaT keratinocyte with *Nemopilema nomurai* jellyfish venom (NnV) to study detailed mechanisms of actions behind the skin symptoms after jellyfish envenomation. Using two-dimensional gel electrophoresis (2-DE) and matrix-assisted laser desorption-ionization time-of-flight mass spectrometry (MALDI-TOF/MS), cellular changes at proteome level were examined. The treatment of NnV resulted in the decrease of HaCaT cell viability in a concentration-dependent manner. Using NnV (at IC_50_), the proteome level alterations were determined at 12 h and 24 h after the venom treatment. Briefly, 70 protein spots with significant quantitative changes were picked from the gels for MALDI-TOF/MS. In total, 44 differentially abundant proteins were successfully identified, among which 19 proteins were increased, whereas 25 proteins were decreased in the abundance levels comparing with their respective control spots. DAPs involved in cell survival and development (e.g., Plasminogen, Vinculin, EMILIN-1, Basonuclin2, Focal adhesion kinase 1, FAM83B, Peroxisome proliferator-activated receptor-gamma co-activator 1-alpha) decreased their expression, whereas stress or immune response-related proteins (e.g., Toll-like receptor 4, Aminopeptidase N, MKL/Myocardin-like protein 1, hypoxia up-regulated protein 1, Heat shock protein 105 kDa, Ephrin type-A receptor 1, with some protease (or peptidase) enzymes) were up-regulated. In conclusion, the present findings may exhibit some possible key players during skin damage and suggest therapeutic strategies for preventing jellyfish envenomation.

## 1. Introduction

Mainly the jellyfish number has expeditiously proliferated in current years; thus, jellyfish envenomation cases shoot up. According to the National Institutes of Health (NIH), 150 million people were accidentally injured by jellyfish every year. The jellyfish *Nemopilema nomurai* from phylum cnidaria is around 2 m in diameter and 200 kg body weight, is one of the giant jellyfish in the world [1]. It has special stinging cells (nematocysts) attached to several thread-like tentacles. NnV is the fusion of complex components abundant in peptides and proteins, many bioactive proteins having a cytotoxic, hemolytic, hepatotoxic, cardiotoxic activity, or neurotoxic components reside inside hollow tubules in the nematocysts [2,3,4,5]. A significant difference in jellyfish venom compositions between different species was noticed based on their geographical locations, as well as seasonal variations [4].

The cnidarian nematocysts are of various sizes and shapes, produced by special cells called nematocytes, and they are bulb-shaped capsules containing a coiled, hollow, usually barbed thread that forcefully eject outward upon stimulation [6]. Cnidocils are hair-like sensory processes projecting from a cnidoblast’s surface, believed to trigger the discharge of the nematocyst [7]. During contact with human skin or prey, jellyfish tentacles release thousands of toxin-enriched nematocysts into the skin through the shaft, injecting their venomous components simultaneously. Mostly, swelling and itching are the everyday observations in the affected skin area, but sometimes in severe cases, it can cause skin necrosis [8]. Moreover, histamine is released due to the exocytosis of mast cells by jellyfish venom [9]. Jellyfish stings can also result in many systemic symptoms such as neurological gastrointestinal, cardiac, or allergic responses after entering the general circulation [10]. Ventricular arrhythmias and cardiac arrest due to jellyfish venom in severe cases may lead to mortalities [11,12,13]. Acute renal failure was also observed due to intravascular hemolysis by jellyfish envenomation [12]. The box jellyfish *Chironex fleckeri* is the most venomous marine creature, and its envenomation can cause cardiorespiratory distress and leads to death within few minutes [14,15]. Jellyfish *Chironex fleckeri* toxins can induce edema, vesicle formation, erythema, result in extensive progression of necrosis and cause purple to brown wounds [14,15,16]. The everlasting serious impediment of such wounds consists of granulomas, hyperpigmentation, fat atrophy, and keloids [15,16].

Dermal toxicity is prevalent in response to venom from different organisms, and the mode of action is the same. Earlier it was demonstrated that snake venom protease could cause edema, hemorrhagic activity, inflammatory response, and necrosis in injured skin [17]. *N. annulifera* snake venom contains multiple components responsible for local and systemic inflammatory reactions, disrupt the human coagulation system, cause lung hemorrhage, ultimately leads to death, and venom is cytotoxic human keratinocytes [17,18,19]. It was demonstrated earlier that sea urchin envenomation could cause mast cell degranulation, disruption of cell metabolism, cause frequent pain, bleeding, edema, dermatitis, cardiovascular collapse, and respiratory failure [20].

Previously, it was reported that American scorpion *Didymocentrus krausi* result in thrombotic effect and fibrinogenolytic activity in vivo and in vitro animal models [21]. An earlier study suggested that *Palythoa zoanthid* corals and *Ostreopsis dinoflagellates*-derived toxin named palytoxin (PLTX) had caused deleterious effects on human health [22]. PLTX exposure resulted in severe dermatological distress during Ostreopsis blooms; hence PLTX triggers the cytotoxic effects in vitro [22]. Our former investigation has shown that the metalloproteinase present in the scyphozoan jellyfish venom predominately mediates dermal toxicity [23,24]. However, the mechanism of dermal toxicity involving NnV effects remain unexplored. NnV is the fusion of complex components abundant in peptides and proteins, which are discharged after proper provocation [25].

Our recent proteomics studies revealed that *Nemopilema nomurai* venom contains various novel components such as phospholipase D Li Sic Tox beta IDI, a serine protease, putative Kunitz-type serine protease inhibitor, phospholipase A2, disintegrin and metalloproteinase, leukotoxin, hemolysin, three-finger toxin MALT0044C, allergens, venom prothrombin activator trocarin D, tripeptide Gsp 9.1, and many other toxin proteins [25]. Moreover, other transcriptomics, genomics, and proteomics studies have recognized numerous classes of toxins from *NnV*, comprising metalloprotease, hemolysin, cytotoxins, c-type lectin, potassium channel inhibitor, g channel-forming toxins, thrombin, and many others [14,15]. Hence have a fatal role in the lethality caused by NnV stinging.

Such components may contribute to venom toxicity and leads to the deleterious effect of venom on human health. The molecular mechanism behind the jellyfish’s envenomation on human skin cells is still unidentified. In the current research, we have revealed *Nemopilema nomurai* jellyfish venom’s critical targets in human skin cells using HaCaT model keratinocytes.

## 2. Results

### 2.1. NnV Induces Cytotoxicity in HaCaT Cell Lines

The dermal toxicity effect of NnV on HaCaT cells was analyzed using MTT (3-(4,5-dimethyl-2-yl)-2,5-diphenyltetrazolium bromide) assay. In short, cumulative concentrations (0–5 μg/mL) of NnV were used to treat the HaCaT cells for 24 h. Cells without NnV treatment were considered control, NnV treatment reduced the growth of HaCaT cells in a concentration- and time-dependent manner (Figure 1). On treatment with the lower concentrations (0.5 and 0.8 μg/mL) of NnV there was no substantial change observed in cell viability after 24 h of treatment. While with 1, 3, and 5 μg/mL NnV, Cell growth was reduced to 57.3%, 47.1%, and 35.4%, respectively. The cell viability was significantly reduced after treatment with 3 μg/mL of NnV for 24 h in comparison to the control cells. Probit analysis was used to determine the IC50 value of NnV concentration (2.5 μg/mL); at this concentration, a 50% reduction in cell proliferation was observed. Further, the number of viable cells was significantly decreased after NnV treatment in a concentration- and time-dependent manner and NnV had antiproliferative activity. The IC50 value (2.5 μg/mL) of NnV was used for treating HaCaT cells for proteomic analysis. The morphological characteristics of HaCaT cells were analyzed under the phase-contrast microscopic (Figure 2). In the present study, we have evaluated that NnV displayed cytotoxic effects in HaCaT cells at different periods (12 h and 24 h). Compared to the control, the NnV treated cells showed loss of adhesion with round shape, shrinking diameter, necrosis, and development of apoptotic fragments and decreased percentage of viable cells (Figure 2). Cell viability was rapidly decreased in 12 h, and 24 h NnV treated HaCaT cells compared to untreated control cells. NnV treated HaCaT cells after 24 h showed a significant decrease in cell viability as compared to 12 h treated cells. In contrast, HaCaT control cells shown average growth. DAPI staining assay was performed to determine the effect of NnV on apoptosis, nuclear morphological change in treated HaCaT cells (Figure 3). It was observed that percentage of viable cells decreased in 12 h and 24 h NnV treated HaCaT cells. Cell necrosis and nuclear morphological changes were noticed in venom treated HaCaT cells after 12 h and 24 h.

### 2.2. Two-Dimensional Gel Electrophoresis of Cellular Proteins from NnV-Treated HaCaT Cells

To explore the proteomic changes in HaCaT cells with and without NnV treatment, we performed the relative proteomic study. The protein level variations between non-treated and NnV treated HaCaT at IC50 for 12 or 24 h were assessed by 2-DE. Three independent biological replicates and corresponding gels were run for the analysis from each treatment. After silver staining, around 1000 protein spots were visible. All three gels from the three independent experiments for each treatment showed similar patterns of 2-DE protein spots supporting the reproducibility of our experiment. Figure 4 shows the representative 2-DE images of protein isolated from non-treated and NnV-treated (12 h and 24 h) HaCaT cells. After scanning all 2-DE gels, Progenesis Same software (Nonlinear Dynamics, New Castle, UK) was used to analyze the images. Seventy differentially abundant protein spots with more than a 1.5-fold change in intensity and statistical significance (ANOVA *p*-value_0.05) were picked after automatic spot detection and image analysis. The 2-DE gel image was generated after fusing the control’s images, and NnV treated samples (12 h and 24 h) (Figure 5). MALDI/TOF/MS analysis was performed to determine these 70 differentially abundant proteins.

### 2.3. Ontological Classification of Differentially Abundant Proteins

Based on the gene ontology classification, the 44 identified proteins which were significantly increased or decreased in NnV treated HaCaT cells were further classified based on the molecular function, biological process, protein class, and cellular component of the molecules (Figure 6). Based on molecular component ontology, the proteins’ two main functional categories are binding (43.30%) and catalytic activity (26.70%). Some proteins exhibited molecular function regulator (13.30%), transporter activity, and transcription regulator activity (6.70%). Most proteins were associated with biological regulation (23.1%), cellular component organization or biogenesis (17.9%), metabolic process (15.4%), developmental process and cellular process (10.30%), biological adhesion and localization (5.10%), and multicellular organismal process (2.60%) based on their biological processes. Later these proteins were classified based upon functional protein classes; the dominant class is of transcription factor and enzyme modulator (22.7%); the second abundant class is hydrolase (18.20%). Several proteins share these categories: cytoskeletal protein (9.1%), transporter, transfer/carrier protein, transferase, nucleic acid binding, and calcium-binding protein (4.50%), as shown in the figure. In terms of cellular components, the majority of these proteins were localized in cell and organelle (32.3%), membrane (16.10%), protein-containing complex (9.70%, cell junction, and synapse (3.2%), as shown in Figure 6. Identified proteins belong to vital signaling pathways like apoptosis signaling pathway, histamine H1 receptor-mediated signaling pathway, angiogenesis, Alzheimer disease-amyloid secretase pathway, Integrin signaling pathway, alpha-adrenergic receptor signaling pathway, inflammation mediated by chemokine and cytokine signaling pathway, endothelin signaling pathway, EGF receptor signaling pathway, gonadotropin-releasing hormone receptor pathway, thyrotropin-releasing hormone receptor signaling pathway, oxytocin receptor-mediated signaling pathway and blood coagulation shown in Figure 7.

### 2.4. Protein-Protein Interactions

String interactions illustrate protein-protein interactions among the up and down-regulated proteins, as shown in Figure 7. Most of the proteins were classified into three distinct groups. All the groups were interconnected with each other. The first group consists of focal adhesion kinase (PTK2), vinculin (VCL), which is involved in cell-matrix adhesion, cell proliferation, cell spreading maintain cell morphology and locomotion. VCL is present in the middle of the group, displays interaction with other essential proteins such as α-Actinin-4 and PTK2. The bioinformatic analysis suggested that Toll-like receptor 4 participated as an essential key player with plasminogen (PLG) and glucocorticoid receptors (NR3C1), which plays a crucial function in epidermal development, cell migration, proliferation, and wound healing. The KEGG (Kyoto Encyclopedia of Genes and Genomes) pathway and the string interactions for individual proteins are shown in Table 1.

## 3. Discussion

In the coastal areas, jellyfish envenomation cases are on the rise worldwide. According to the National Institutes of Health (NIH), around 150 million people accidentally get injured by jellyfish every year [1]. Jellyfish envenomation can induce local consequences such as erythematous eruption, flaring sensations, edema, and necrosis and generate critical systematic symptoms including cardiovascular discomfort, respiratory impairment and hypovolemic attack [2,3,4,5]. From our prior findings, we have found that most of the scyphozoan jellyfish venoms are rich in metalloproteinase ingredients, which have a vital role in the pathogenesis of damaged skin tissue in an in vivo model [23,24]. The jellyfish *Nemopilema nomurai* induce life-threatening effects in human beings and cause many deleterious effects [2]. Our earlier proteomics analysis provides an overall complete understanding of the NnV components, which are described above [25]. Many investigators have examined dermal toxicity in other venomous organisms and have similar means of envenomation. Earlier studies suggested that Brown spiders (*Loxosceles* genus) venom comprised toxins supplemented with metalloproteinase, phospholipase D, hyaluronidase, and phosphatase [26]. Brown spider accidental bites can cause skin degeneration, intense inflammatory reaction, platelet aggregation, intravascular hemolysis, and cause thrombocytopenia. Phospholipase D is a dermonecrotic toxin that generates necrotic lesions, platelet aggregation, hemolysis, cell cytotoxicity, nephrotoxicity, enhanced vessel permeability, and death in animals [26,27]. According to earlier research, a novel membrane disrupting toxin identified in *Vespa mandaria* venom can cause intense pain, dermal necrosis, edema, and tissue damage [28]. For the present study, MALDI/TOF/MS was used to identify the differentially abundant proteins upon the treatment of NnV. Among 44 differentially expressed identified proteins, 25 proteins shown diminished volume, whereas 19 proteins exhibited an increment in volume after NnV treatment. Identified proteins are shown in Table 2. The fold change and *p*-value of these identified proteins are summed up in the Appendix A. The representative MALDI/TOF/MS spectra of 4 identified proteins of interest are provided in Appendix A, such as Vinculin, Plasminogen, Emilin, and Focal adhesion kinase 1.

### 3.1. Peroxisome Proliferator-Activated Receptor γ Coactivator 1α (PGC-1α)

Our proteomic result illustrated that the protein level of Peroxisome proliferator-activated receptor γ coactivator 1α (PGC-1α) was reduced after venom treatment. PGC-1α is a transcriptional coactivator that regulates the expression of antioxidant genes to modulate the oxidative stress response [29]. The class of 209 known environmental pollutants represents Polychlorinated biphenyls (PCBs) [29]. Monochlorobiphenyl (PCB3) is a semi-volatile PCB congener whose traces were found in human blood samples, commercial paints, and the polluted environment. 1-(4-Chlorophenyl)-benzo-2,5-quinone (4-ClBQ) is a metabolite of 4-monochlorobiphenyl, PCB3. Earlier studies showed that 4-ClBQ exposure impedes the activity of PGC-1α [29] and expression of catalase, which directly associates with an upsurge in cellular reacted oxygen species (ROS) levels and toxicity in HaCaT cells [29]. PGC-1α is also down regulated after the treatment with NnV, leading to toxicity in HaCaT cells.

### 3.2. Elastin Microfibril Interface Located Protein 1 (EMILIN-1)

EMILIN’s belong to the protein family dominant in the extracellular matrix and possess a distinctive structural domain configuration. Until now, four such genes were known in humans and mice [30]. EMILIN-1 (Elastin microfibril interface located protein 1) comprises N terminal cysteine-rich EMI domain, coiled-coil alpha-helical domain, collagenous domain, and qc1qlike domain [30]. EMILIN-1 is a multifunctional protein involved in cell migration, proliferation, and cell adhesion. Emilin-1 is found in various connective tissues, e.g., lung and blood vessels [31]. EMILIN-1 is also present in the dermis and plays an important role in maintaining the 3D structure of the extracellular matrix [32]. Earlier studies suggested that antibodies counter to EMILIN-1 repressed elastin deposition by smooth muscle cells (in vitro) and proposed that this protein may lead to elastogenesis [30,31]. EMILIN-1 promotes vascular cell maintenance by steadying molecular interactions between elastic fiber components and providing specific cell adhesion properties to elastic fibers [30,31,32,33,34]. Earlier studies suggested that the absence of EMILIN-1 resulted in elastogenesis and vascular cell defects [34]. Our results showed that EMILIN-1 level was decreased after NnV treatment and hinted toward the severe damage caused by NnV to HaCaT cells.

### 3.3. Basonuclin 2

Basonuclin 2 is a transcription factor that belongs to a member of the Basonuclin zinc-finger family proteins present in germ tissues, corneal epithelium, basal skin keratino- cytes stratified squamous epithelium and also expressed in hair follicles keratinocytes [35]. It acts as a transcription factor and generates nuclear function by regulating gene expression in different types of cells [35]. It plays a crucial role in the growth of craniofacial bones and male germ cells. The lack of Basonuclin 2 gene expression in mice leads to neonatal lethality correlated with cleft palate and craniofacial deformity. Thus, the reduced activity of Basonuclin 2 can result in human craniofacial abnormalities. Earlier research suggested that Basonuclin 2 is vital for typical mitotic arrest, prevention of premature meiotic initiation, and meiotic progression in mouse male germ cells [35,36]. Basonuclin 2 also plays an essential role in the multiplication of embryonic craniofacial mesenchymal cells and is orthologous to disco proteins [36,37,38]. The current studies evaluated Basonuclin 2 protein was decreased in abundance in NnV treated HaCaT cell after 12 h and 24 h treatment. A reduced amount of Basonuclin 2 supports the severe NnV induced toxicity in HaCaT cells.

### 3.4. Glucocorticoid (GC)

Glucocorticoid (GC) derivatives are the utmost efficient and extensively referred compounds for curing inflammatory and autoimmune diseases [39]. GC deficiency can cause Addison’s disease and result in skin alterations, suggesting that proper GC levels are essential for regular tissue function [39,40,41,42]. Former studies indicated that epidermal depletion of GR disrupts the epidermal barrier by augmented proliferation and impaired differentiation [40,41,42,43]. It was shown earlier that epidermal GR and mineralocorticoid receptor [MR] work mutually to control epidermal development and respond to skin inflammation [42]. Similarly, in the current case, it might be concluded that the low abundance of glucocorticoid (GC) may also lead to dermal toxicity in NnV treated HaCaT cells.

### 3.5. Plasminogen

Ubiquitous plasminogen activates Plasmin by the action of cell-derived urokinase-type (uPA) or tissue-type (tPA) plasminogen activator [44]. Plasmin is a fibrinolytic protease. Earlier studies suggested that uPA mediated plasminogen activation contributes to the process of human skin wound healing [44]. Henceforth plasminogen activation system signifies an essential molecular mechanism of extracellular protein degradation and is an effective component of typical wound healing. The plasminogen activation system significantly contributes to tissue remodeling, and tissue-type plasminogen activator (tPA) primarily deals with intravascular fibrinolysis [44,45,46]. During inflammation plasminogen activator system is engaged in complement activation, inflammatory cell migration, and cell signaling. Plasminogen (plg) deficient and wild-type (wt) mice models were used to investigate the function of plg in cutaneous wound healing [44,45,46]. It was found that plg deficient mice showed delayed wound healing and inflammation augmentation after two months of injury. It was also reported that when plg deficient mice were injected intravenously with human plg, it showed declined inflammation and alleviation in the wound healing [46,47]. All of the evidence resolutely indicates that plasminogen is a promising drug substitute to treat non-healing chronic wounds. It was reported that plasminogen activator inhibitor-1 causes impaired wound healing in venous leg ulcers, and the level of tissue-type plasminogen activator (tPA) expression is changed in keratinocytes at the site of injury [47]. Our proteomic results showed that plasminogen volume is declined after NnV treatment, proposing a probable mechanism of actions for NnV-induced dermal toxicity.

### 3.6. Vinculin

An Actin filament (F-actin)-binding protein Vinculin plays a vital role in cell-matrix adhesion and cell-cell adhesion. Vinculin modulates the expression of cell-surface E-cadherin and accelerates the mechanosensing by the E-cadherin complex [48]. Vinculin controls cell signaling and affects cell contractility and adhesion-site turnover [48,49]. An inadequate amount of Vinculin in cells causes a deformity in cell migration, cell spreading, cell-matrix adhesion, and focal adhesion turnover. Earlier research revealed that the downregulation of Vinculin in atherosclerotic plaque leads to disorganized tissue, increase sensitivity to mechanical injury, and delayed healing processes [49]. Vinculin sh RNA knockdown/substitution model system notably proposed that the deprivation of vinculin cause deterioration of epithelial cell-cell adhesion due to the reduction in cell surface expression of E. cadherin [48,49]. We also observed a decrease in the expression of Vinculin after NnV treatment.

### 3.7. Focal Adhesion Kinase 1

Focal adhesion kinase 1 plays a crucial role in modulating cell adhesion, spreading, migration, actin cytoskeleton reorganization, formation and disassembly of focal adhesions and cell protrusions, cell cycle progression, apoptosis and cell proliferation [50,51]. It was reported earlier that focal adhesion kinase functions upstream of phosphatidylinositol 3-kinase/Akt. It regulates fibroblast survival in retort to shrinkage of Type I collagen matrices by an integrin Beta 1-viability signaling pathway [51]. It was observed earlier that focal adhesion Kinase 1 triggers the migration and proliferation of keratinocytes at epidermal injury site [52]. Thus, focal adhesion Kinase 1 seems to have a crucial role in the reepithelization of human wounds. Our data show a decrement in Focal adhesion Kinase 1 volume in response to NnV, suggesting that NnV can affect cell cytoskeleton and cause dermal toxicity with compulsive disorders.

### 3.8. Myocardin Related Transcription Factor-A (MRTF-A)

A transcriptional coactivator named Myocardin related transcription factor-A (MRTF-A) is known to be dominantly localized in the cytoplasm and but during mechanical stress or growth factor stimulation, it translocates to the nucleus [53]. The activity of MRTF-A increase in a time-dependent manner during scleroderma (systemic sclerosis, SSc), and lower expression of MRTF-A or any hindrance in nuclear translocation may affect the development of fibrosis in SSc skin [54]. The basal skin and lung stiffness were reduced in the MRTF-A null mouse due to changes in fibrillar collagen. MRTF-A plays a vital role in SSc fibrosis, and it remodels the extracellular matrix in response to mechanical signals [55]. Increased protein level of MRTF-A in HaCaT cells post-NnV treatment clues severely damaged human keratinocytes by NnV.

### 3.9. Toll-Like Receptors (TLRs)

A membrane-localized pattern recognition receptor named Toll-like receptors (TLRs) is a component of the endogenous immune system. TLRs participate in the signal transduction pathway involved in response to inflammation and plays a significant role in the endogenous immune response [56,57,58]. Exposure to solar UVB irradiation may lead to sunburn, erythema, oxidative stress and apoptosis, and activation of various signaling pathways, including the toll-like receptor (TLR)4 [58]. It suggests that (TLR)4 is crucial in the progression of inflammatory response and pathogenesis of inflammatory skin diseases. The level of the toll-like receptor (TLR)4 is high after NnV treatment, proposing a probable mechanism of action for NnV-induced dermal toxicity in HaCaT cells.

### 3.10. Minichromosome Maintenance (MCM)

The Minichromosome maintenance (MCM) is cycling cell-specific initiation factors for DNA replication [59] which dominantly express during actinic keratosis (AK) in atypical keratinocytes [60]. AK is a precancerous lesion of the skin with variable rates of transformation into non-melanocytic carcinomas [61]. MCM 2 protein acts as a diagnostic marker for AK [61]. Our data shows elevation in MCM 2 protein level in response to venom treatment, suggesting that NnV can cause dermal toxicity and severe pathological disorder.

### 3.11. APN/CD13 (Aminopeptidase N)

APN/CD13 (Aminopeptidase N) is a zinc-dependent transmembrane ectopeptidase, it mainly split neutral amino acids from the N-terminus of peptides. Besides, APN/CD13 is also involved in extracellular matrix (ECM) degradation [62,63]. It was previously reported that APN/CD13 express in dermal, gingival fibroblasts, and dermatological/rheumatic diseases derived human fibroblasts [63,64]. Psoriasis vulgaris is a well-known skin inflammatory disease that leads to scaly well-demarcated skin plaques accompanied by a burning sensation. In psoriatic fibroblasts, APN/CD13 is over-expressed and showed higher activity as compared with HD fibroblasts [63,64]. Our results are in conjunction with previous findings, which also shown a significant increase in APN/CD13 (Aminopeptidase N) protein volume in NnV treated HaCaT cells.

## 4. Conclusions

In conclusion, NnV consists of various ingredients, which are ample in toxin proteins and peptides, cause local and systematic inflammatory responses after accidental stinging. In current study NnV has altered the pathways related to immune response, cell adhesion, actin cytoskeleton, apoptosis, cell cycle, blood coagulation. Our previous studies explored that metalloproteinase, phospholipase, and other toxins components result in pathogenesis generated by the NnV after stings. From protein interaction network analysis, we have found that focal adhesion kinase (PTK2) is in the center of network and upstream of Vinculin (VCL) and α-Actinin-4. Focal adhesion kinase (PTK2) level was decreased after NnV treatment which leads to decreased abundance of VCL and α-Actinin-4. All these proteins have an influential role in cell-matrix adhesion, spreading, migration, actin cytoskeleton reorganization, formation and disassembly of focal adhesions and cell protrusions, cell cycle progression, apoptosis and cell proliferation. Further extensive studies are needed to demonstrate the detailed mechanism and role of these target candidate proteins in the dermal toxicity caused by NnV. Mainly toll-like receptor (TLR)4 plays a central role in signal transduction pathway, cause progression of inflammation and pathogenesis of skin diseases. However, we have evaluated that TLR4 level was increased in response to NnV treatment, which hints towards its role in sensing and relaying dermal toxicity and skin injury. Such a target-based study in the future could explore therapeutic care against Jellyfish envenomation. NnV inheres multitudinous enzymatic ingredients may significantly categorize promising bioactive components of NnV, which leads to dermal toxicity. For the present study, we have focused on analyzed the proteome profiling to characterize NnV-induced dermal cell cytotoxicity.

## 5. Remarks

It is presumed that the present research may have imparted awareness dealing with NnV envenomation, their management, therapy and may provide an invaluable understanding of pernicious consequences of the NnV stinging.

## 6. Materials and Methods

### 6.1. Chemicals and Reagents

The chemicals used and manufacturer details are as follows: the sequencing grade modified trypsin (Promega, Corporation, Madison, WI, USA), acetonitrile (ACN), trifluoroacetic acid (Merck chemicals, Darmstadt, Germany), formic acid (Acros Organics BVBA, Geel, Belgium), Immobiline^TM^ Dry strip (pH 4–7, 18 cm), and iodoacetamide were bought from GE Healthcare life sciences (Marlborough, MA, USA). Most of other chemicals and reagents with analytical grade were purchased from Sigma Aldrich (St. Louis, MO, USA).

### 6.2. Sample Collection and Preparation

Numerous *N. nomurai* jellyfish specimens were collected from the Yellow Sea along the coast of Gunsan, South Korea. Tentacles were separated and transported to the lab instantly to the laboratory under ice-cold conditions for further examination. Nematocysts were isolated using the method as described previously [65]. The dissected tentacles were washed several times with cold seawater to remove salts and debris. To the washed tentacles, three volumes (*v*/*v*) of cold seawater was added and kept on the shaker for 24 h at 4 °C. Subsequently the tentacles free seawater was centrifuged at 1000× *g* for 5 min, the pellet was collected and rinsed three times with seawater. Sedimented tentacles were further autolyzed in fresh seawater at 4 °C for one day as described above, and the autolysis process was repeated for 3–4 days. Finally, steady nematocysts were washed carefully several times with seawater. Nematocysts were collected at 500 g for 5 min, and the supernatant was removed, nematocyst pellets were dried in freeze dryer lyophilizer. Lyophilized pellets were stored at −80 °C until further use.

### 6.3. Venom Extraction and Preparation

Venom was pulled out from the lyophilized-dried nematocysts employing the method designated by Carrette and Seymour [66] with minor modulations. In brief, venom was extracted from 50 mg of nematocyst using glass beads (approximately 8000 beads; 0.5 mm in diameter) and 1 mL of ice-cold phosphate-buffered saline (PBS, pH 7.4). These samples were shaken in a mini bead mill at 3000 rpm for 30 s, repeated ten times with intermittent cooling on ice. The venom extracts were then transferred to a new Eppendorf tube and centrifuged (22,000× *g*) at 4 °C for 30 min. This supernatant was used as NnV for the present study. Bradford assay was explored (Bio-Rad, Hercules, CA, USA) [67] to determine the protein concentration of venom samples.

### 6.4. Cell Culture

The human keratinocytes cell lines HaCaT was obtained from the American Type Culture Collection (ATCC) (Manssas, VA, USA) and carefully cultured in Dulbecco’s modified Eagle’s medium (DMEM) comprising 10% heat-inactivated fetal bovine serum (FBS), 100 µg/mL penicillin-streptomycin-amphotericin B solution. The cells were maintained in a humidified incubator at 37 °C, and with 5% CO_2_, and their medium was replaced every 2–3 days.

### 6.5. Conditions for MTT Assay for Cell Viability

MTT (3-(4,5-dimethyl-2-yl)-2,5-diphenyltetrazolium bromide) reduction assay was used to measure cell viability. For the MTT assay, 24-well plates were used to seed the HaCaT cells at the density of 4 × 10^4^ cells/well, and cells were grown for 24 h. The concentrations ranging from 0–5 µg/mL of NnV were used to treat the cells for the next 24 h. After 24 h incubation MTT dye 50 µL (0–5 mg/mL) was added to the cells in each well, and then cells were further incubated at 37 °C for the next 3 h. The supernatant was removed and replaced by 200 µL of dimethyl sulfoxide (DMSO) in each well, and to solubilize the formazan salts produced during reaction plates were kept on the shaker for 10 min. GENios microplate spectrophotometer (PowerWave^TM^XS, BioTek Instruments, Inc., Winooski, VT, USA) was used to measure the absorbance of samples at 540 nm. Probit analysis [68] was used to quantify the IC50 value. The phase-contrast microscope was used to analyze the morphological changes made by NnV treatment. Untreated cells were used as the control, minimum of three replicates were used for each sample to perform MTT assay.

### 6.6. DAPI Staining for Nuclear Morphological Analysis

The IC_50_ concentration of NnV was used to treat the 24 h grown HaCaT cells (1 × 10^4^ cells/well) plated onto cell culture slide (SPL life science, Pocheon, Korea)for 12 h and 24 h to analyze the nuclear morphology. The cells on the slide were fixed with the 4% paraformaldehyde treatment for 15 min. Followed by three washings each for 10 min. Then cells were stained with DAPI solution, and images were taken under fluorescence microscopy.

### 6.7. Protein Extraction and Sample Preparation

The 2-DE protein samples were prepared from the HaCaT cells at 70% confluence were treated with IC_50_ concentration of NnV for 12 and 24 h. The ice-cold 1X phosphate-buffered saline (PBS) was used to wash treated cells and 2-DE lysis buffer (7 M Urea, 2 M thiourea, 4% CHAPS (3-[(3-cholamidopropyl) dimethylammonio]-1 propane sulfonate, 10% DTT, 0.5% IPG buffer with 1% proteinase inhibitor was used to scrap the cells. Samples were vortexed for 30 min at 4 °C and the insoluble parts were removed by centrifugation (14,000× *g*, 15 min, 4 °C). To cell lysis supernatant, an equal volume of 20% TCA was added, the mixture was incubated for 30 min on ice and centrifuged at 14,000× *g* for 15 min at 4 °C. The supernatant was removed, and pellets were washed twice with 200 µL of ice-cold acetone and centrifuged as above. Then the pellets were vacuum dried for 10–15 min. Subsequently, dried pellets were dissolved in a sample buffer containing (7 M urea, 2 M thiourea, 4% (*w*/*v*) CHAPS). The protein concentration of the samples was determined by using the Bradford assay.

### 6.8. Two-Dimensional Gel Electrophoresis and Image Analysis

The 2-DE was used to examined the differentially abundant proteins. A total of 300 µg of protein samples were resuspended in 340 µL of a rehydration buffer comprising (2 M thiourea, 7 M urea, 4% (*w*/*v*) CHAPS, 10 mg/mL DTT, 1% pharmalytes 4–7 and low amount of bromophenol blue). For the rehydration process, protein samples were loaded into the rehydration try. The 18 cm immobiline TM Dry strip (pH 4–7), were placed carefully on the rehydration try for overnight at room temperature. Through Ettan IPGphor system (GE Healthcare, Salt Lake City, UT, USA). Isoelectric focusing was achieved and procedure comprises series of subsequent sitting: 50 V for 1:00 h, 200 V for 1:00 h, 500 V for 0:30 h, gradient 4000 V for 0:30 h, 4000 V for 1:00 h, gradient 10,000 V for 1:00 h, 10,000 V for 13:00 h and 50 V 3:00 h at 20 °C. Strips were rinsed with a solution covering 50 mM Tris-HCl (pH 8.8), 6 M urea, 30% glycerol, 2% SDS and 0.01% bromophenol blue containing 1% *w*/*v* DTT for undertaking reduction. Afterward, the reduced strips were wiped in an alkylation solution, integrated with 2.5% *w*/*v* iodoacetamide. After washing, the strips were positioned on the upper side of 18 cm, 12% SDS polyacrylamide gels, and fixed with 0.5% (*w*/*v*) agarose gel, including a bit of bromophenol blue. Concurrently in the second dimension, the proteins were resolved at 20 mA/gel at 20 °C.

For each sample, quadruplicate gels were prepared, and gels were silver stained, utilizing the technique described by Mortz et.al [69]. The image investigation was achieved with Progenesis Same Spots software (Nonlinear Dynamics, Newcastle, UK. For that, the gels were scanned and visualized by the Epson perfection V 700 photo scanner (Epson, Suwa, OU, Japan). The significant protein spots were recognized by editing program (which carried out, spot detection, background diminution, emulated spot, and entire spot volume normalization) to discover the differentially abundant protein spots. The distinctness in expression level between HaCaT cells and HaCaT-NnV treated at a different time was evaluated statistically by one-way ANOVA depend upon value *p* ≤ 0.05.

The analyzed protein spots showing the difference in expression were incised, taken out from 2-DE gels, and in-gel digestion of the proteins was acquired by the former technique [65]. For 10 min, the gel bits were destained in 100 mM Na_2_S_2_O_3_ 30 mM potassium ferricyanide and 100 mM Na_2_S_2_O_3_ (1:1, *v*/*v*). 100% ACN was used to dehydrate the gel bits and dried in a speed vacuum apparatus. Then the dried gels bits were reduced with (10 mM DTT in 100 mM ammonium bicarbonate) reduction solution, incubated at 56 °C for 45 min. Thenceforth the gel bits were alkylated (100 mM iodoacetamide in 100 mM ammonium bicarbonate) alkylation solution, kept in a shaker for 45 min in the dark. After that that the gel bits were incubated in a digestion solution containing 50 mM NH_4_HCO_3_ and 20 ng/µL trypsin on ice. After 45 min trypsin was taken out, a suitable amount of 50 mM NH_4_HCO_3_ was added, and gel bits Eppendorf were kept overnight at 37 °C. The tryptic peptide mixture was extracted with an extraction buffer containing 100% ACN and 50% TFA and concentrated in a vacuum centrifuge.

### 6.9. MALDI-TOF/MS Analysis and Database Searching

HCCAs matrix solution (α-acyano-4-hydroxycinnamic acid) 1 µL and 1 µL of extraction buffer were used to re-dissolve the peptide extract and spotted onto a clean MALDI-TOF plate. Voyager-DE STR mass spectrometer (Applied Biosystems, Franklin Lakes, NJ, USA) was used to record the mass spectrometry (MS) spectra. The reflection/delayed extraction mode was used to acquire the spectra. The spectra over a mass range of 800–3000 Da were recorded. Monoisotopic peptide masses over a mass range of 800–3000 Da were further analyzed. Protein identification was carried out by peptide mass fingerprinting (PMF) using the MS-Fit program (http://prospector.ucsf.edu (accessed on 5 October 2018) and Mascot (Matrix science http://www.matrixscience.com (accessed on 10 November 2018) in the Swiss-Prot databases. The following parameters were considered for the peptide search; carbamidomethylation of cysteines as a fixed modification, oxidation of methionine as variable modification, peptide mass tolerance of ±10–70 ppm for the fragment ions, trypsin with one missed cleavage was allowed. The number of the matched peptide, the extent of sequence coverage, and probability-based Mowse score was considered before accepting the identification.

### 6.10. Statistical and Bioinformatics Analysis of Protein Identified by MALDI-TOF/MS

One way analysis of variance (ANOVA) was used for the statistical analysis of differentially expressed proteins to evaluate the significance of the difference between the two mean values. *p* < 0.05 and *p* < 0.01 were taken as statistically significant. The Panther classification system (http:www.pantherdb.org/ (accessed on 19 January 2019) was used to classify data in terms of molecular function, biological process, protein class, and cellular components [70].

## Figures and Tables

**Figure 1 toxins-13-00311-f001:**
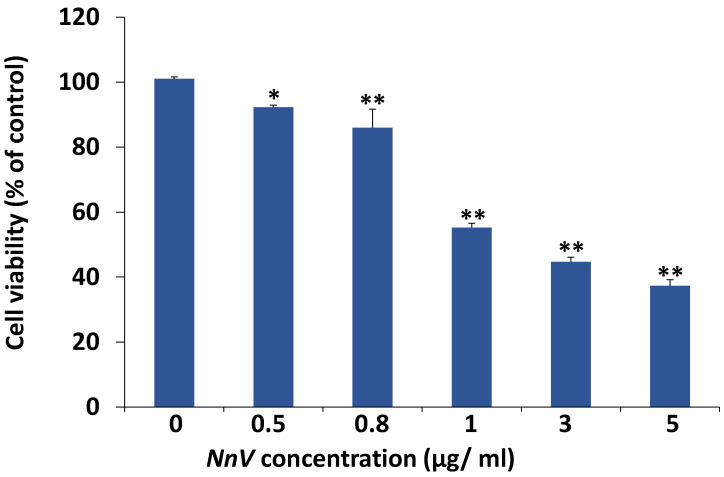
The proliferation of HaCaT human keratinocytes was inhibited by *Nemopilema nomurai* jellyfish venom (NnV). HaCaT cells were exposed to different concentrations of NnV for 24 h, and the 3-(4,5-dimethyl-2-yl)-2,5-diphenyltetrazolium bromide (MTT) Assay was performed to analyze the cell viability. Bar graphs represent the mean ± SD of three independent experiments. The * asterisk specifies a statistically significant difference in comparison to control * *p* < 0.05, ** *p* < 0.01.

**Figure 2 toxins-13-00311-f002:**
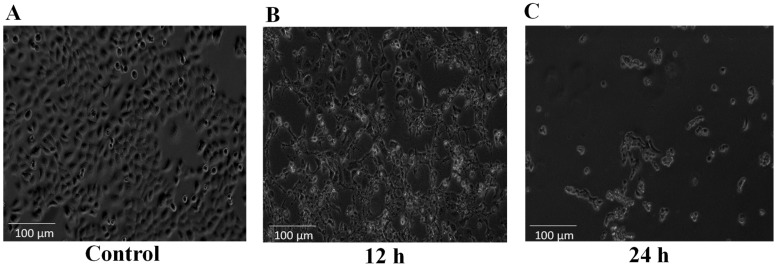
Morphological changes in HaCaT cells exposed to *N. nomurai* jellyfish venom. Treatment with *N. nomurai* jellyfish venom at a concentration of 2.5 µg/mL for 0 h (**A**), 12 h (**B**), and 24 h (**C**). The morphological changes were analyzed under a phase-contrast microscope. Cell viabilities were rapidly decreased in HaCaT cells treated with NnV for 12 h and 24 h compared to untreated control cells.

**Figure 3 toxins-13-00311-f003:**
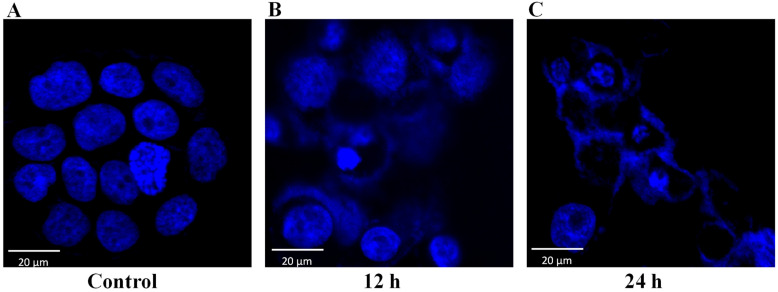
Nuclear staining by DAPI of human HaCaT cells exposed to *N. nomurai* jellyfish venom. HaCaT cells were treated with *N. nomurai* jellyfish venom for 0 h (**A**), 12 h (**B**), and 24 h (**C**). Various morphological changes were seen in the HaCaT cells exposed to NnV, including apoptosis, nuclear condensation, and formation of the apoptotic bodies. The Images shown are representative pictures photographed during the experiment using a fluorescence microscope.

**Figure 4 toxins-13-00311-f004:**
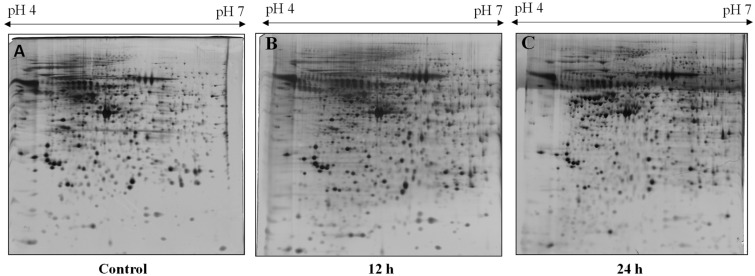
Proteomic comparison of 2-DE images of HaCaT cells treated with NnV (2.5 µg/mL) for 0 h (**A**), 12 h (**B**), and 24 h (**C**). A total of 300 µg protein was resolved on 18 cm IPG dry strips (pH 4–7 L) for the first dimension. Then, 12% SDS-PAGE gels were used for the second dimension. The 2-DE gels were silver stained and scanned by Epson perfection V 700 photo scanner to assess each spot’s protein amount. Three independent experiments for each treatment were carried out to run three 2-DE gels for each treatment.

**Figure 5 toxins-13-00311-f005:**
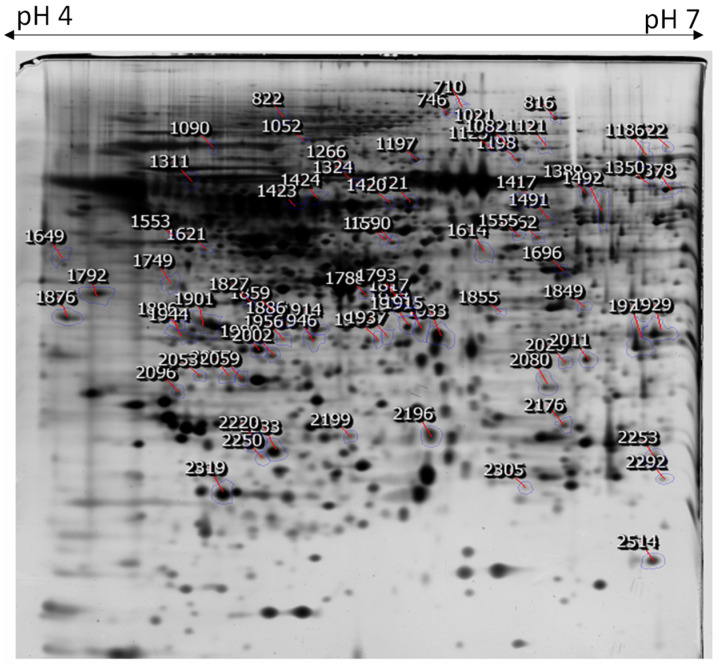
Image showing proteomic analysis of 2-DE of *NnV* treated HaCaT cells. Representative 2D image was generated using Progenesis Same Spots software, and MALDI-TOF/MS was used to analyze the protein spots. The position of differentially abundant proteins were marked with boundaries and arrows.

**Figure 6 toxins-13-00311-f006:**
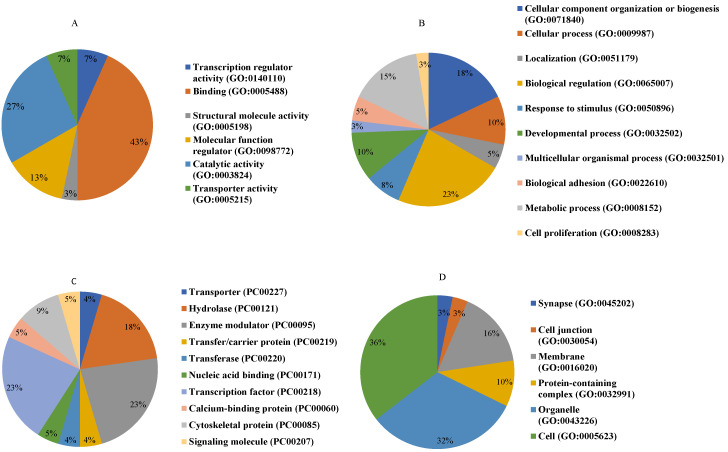
Gene ontology (GO) classification of identified proteins. The proteins were characterized into four groups based on (**A**), molecular function (**B**), biological process (**C**), protein class (**D**), and cellular component. The Panther classification system was used for GO analysis [http:www.pantherdb.org/ (accessed on 3 January 2019)].

**Figure 7 toxins-13-00311-f007:**
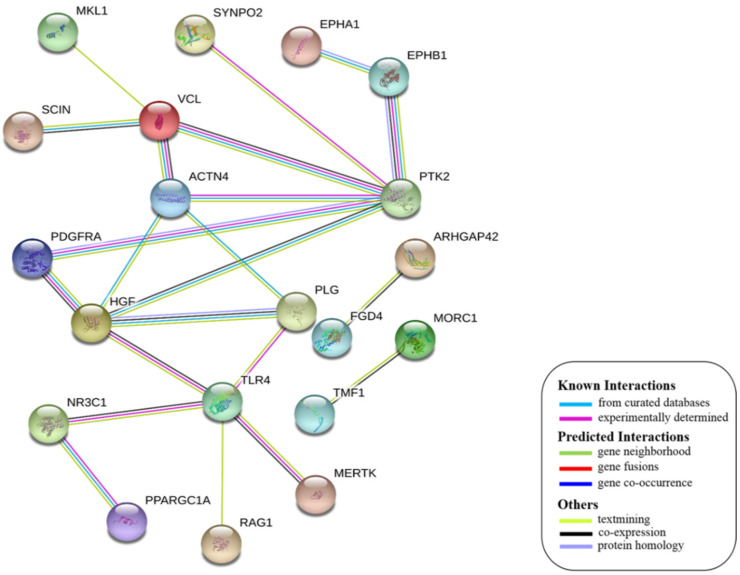
A putative interaction network of known proteins with altered protein amounts using STRING database V 10.5 (http://string-db.org (accessed on 3 January 2019). The basis of interactions is signified by different colored lines, as shown on the right panel.

**Table 1 toxins-13-00311-t001:** The string network interactions of differentially proteins in NnV treated HaCaT cells along with KEGG pathways.

S. No	Protein Name	String Interactions	KEGG Pathways
1	Elastin microfibril interface located protein 1 (EMILIN-1)	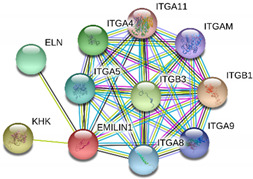	Regulation of actin cytoskeleton, focal adhesion, cell adhesion molecules, PI3K-AKT Signaling Pathway
2	Glucocorticoid	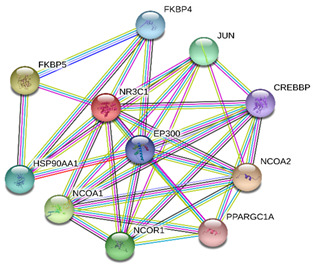	HIF signaling pathway, Thyroid hormone signaling pathway
3	Plasminogen	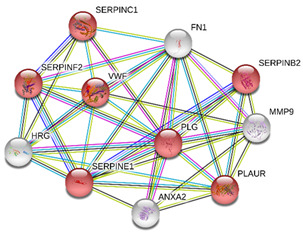	Complement and coagulation cascades
4	Vinculin	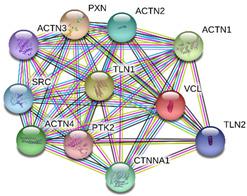	Focal adhesion, leukocyte trans-endothelial migration, regulation of actin cytoskeleton, adherens junction
5	Focal adhesion kinase 1	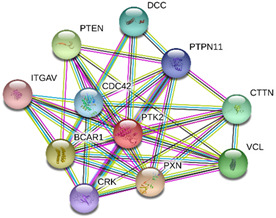	Focal adhesion, bacterial invasion of epithelial cells, regulation of actin cytoskeleton
6	Myocardin Related Transcription Factor-A	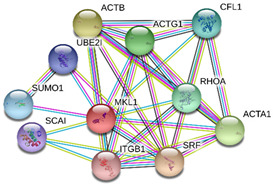	Platelet activation, focal adhesion, tight junction, RAP1 signaling pathway
7	Toll-like receptor 4	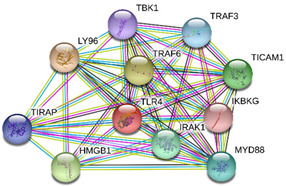	Toll-like receptor signaling pathway, NF- kappa B signaling pathway, pertussis
8	Aminopeptidase N	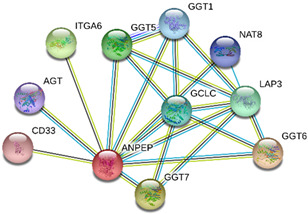	Metabolic pathways, glutathione metabolism, hematopoietic cell lineage
9	DNA replication licensing factor MCM2	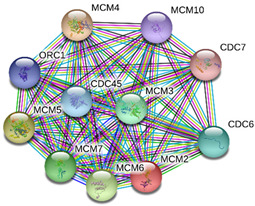	Cell cycle, DNA replication

**Table 2 toxins-13-00311-t002:** Relative abundance of proteins affected by *N. nomurai* venom treatment in HaCaT cells.

Spot No	Accession Number ^1^	Protein Name	Uniprot ID	Theoretical MW/Pi ^2^	Gene	Matched Peptide ^3^	MOWSE Score	Biological Process
Proteins down-regulated by NnV								
2319	P18206	Vinculin	VINC_HUMAN	123,800/5.5	VCL	36.80%	1.64 × 10^14^	epithelial cell-cell adhesion, cell-matrix adhesion
1901	Q9Y6U3	Adseverin	ADSV_HUMAN	80,490/5.5	SCIN	27.30%	6.09 × 10^6^	regulation of chondrocyte differentiation, negative regulation of cell population proliferation
1417	O95259	Potassium voltage-gated channel subfamily H member 1	KCNH1_HUMAN	111,424/7.5	KCNH1	20.10%	20.10%	regulation of cell proliferation, myoblast fusion, potassium ion transport
1989	Q9NQ38	Serine protease inhibitor Kazal-type 5	ISK5_HUMAN	120,716/8.5	SPINK5	19.3%	2.09 × 10^7^	extracellular matrix organization, epidermal cell differentiation, regulation of T cell differentiation
1956	P14210	Hepatocyte growth factor	HGF_HUMAN	83,135/8.2	HGF	28.2%	2.65 × 10^7^	activation of MAPK activity, epithelial to mesenchymal transition, mitotic cell cycle
1937	Q6ZMZ3	Nesprin-3	SYNE3_HUMAN	112,217/5.9	SYNE3	18.60%	9.88 × 10^7^	cytoskeletal anchoring at the nuclear membrane, nuclear migration, regulation of cell shape
1937	P00747	Plasminogen	PLMN_HUMAN	90,570/7.0	PLG	19.30%	2.15 × 10^6^	blood coagulation, fibrinolysis, proteolysis, platelet degranulation
1649	P16234	Platelet-derived growth factor receptor alpha	PGFRA_HUMAN	122,671/5.1	PDGFRA	26.40%	5.01 × 10^8^	double-strand break repair, signal transduction, histone deacetylation
816	O43707	Alpha-actinin-4	ACTN4_HUMAN	104,855/5.3	ACTN4	31.60%	4.53 × 10^9^	actin filament bundle assembly, positive regulation of NIK/NF-kappa B signaling
1052	P54762	Ephrin type-B receptor 1	EPHB1_HUMAN	109,886/6.0	EPHB1	29.4%	9.78 × 10^8^	cell differentiation, cell-substrate adhesion, angiogenesis
1915	O00203	Ubiquitin carboxyl-terminal hydrolase 36	UBP36_HUMAN	122,653/9.7	USP36	14.90%	1.72 × 10^6^	negative regulation of macroautophagy, regulation of protein stability, nucleolus organization
1793	Q9UBK2	Peroxisome proliferator-activated receptor gamma coactivator 1-alpha	PRGC1_HUMAN	91,028/5.7	PPARGC1A	19.00%	3.55 × 10^7^	regulation of transcription, DNA-templated, response to reactive oxygen species, response to ischemia
2057	Q5T0W9	Protein FAM83B	TMF1_HUMAN	114,800/9.0	FAM83B	28.8%	3.02 × 10^12^	cell proliferation, epidermal growth factor receptor signaling pathway
1903	Q6ZN19	Zinc finger protein 841	ZN841_HUMAN	93,149/9.5	ZNF841	23.1%	2.49 × 10^6^	transcription, transcription regulation
1420	Q6ZN30	Zinc finger protein basonuclin-2	BNC2_HUMAN	122,331/6.1	BNC2	27.00%	1.71 × 10^1^	endochondral bone growth, mesenchyme development
1424	Q9Y6C2	Elastin microfibril interface located protein 1 (EMILIN-1)	EMIL1_HUMAN	106,696/5.1	EMILIN1	20.5%	3.88 × 10^7^	cell-matrix adhesion, negative regulation of collagen biosynthetic process
1792	Q9UMS6	Synaptopodin-2	SYNP2_HUMAN	117,515/8.8	SYNPO2	27.3%	9.57 × 10^8^	positive regulation of cell migration, chaperone-mediated autophagy
1423	Q05397	Focal adhesion kinase 1	FAK1_HUMAN	119,234/6.2	PTK2	20.80%	5.88 × 10^6^	positive regulation of cell migration, chaperone-mediated autophagy
1266	Q12866	Tyrosine-protein kinase Mer	MERTK_HUMAN	110,250/5.5	MERTK	16.90%	2.50 × 10^8^	apoptotic cell clearance, cell differentiation, phagocytosis
1933	Q96NW4	Mismatch repair endonuclease PMS2	ANR27_HUMAN	116,985/6.4	ANKRD27	33.90%	2.92 × 10^12^	mismatch repair
822	P04150	Glucocorticoid receptor	GCR_HUMAN	85,660/6.0	NR3C1	32.60%	7.99 × 10^11^	cellular response to glucocorticoid stimulus, signal transduction
2233	Q96M96	FYVE, RhoGEF and PH domain-containing protein 4	FGD4_HUMAN	86,627/5.8	FGD4	16.10%	9.69 × 10^6^	actin cytoskeleton organization, regulation of cell shape, cytoskeleton organization
1944	Q13342	Nuclear body protein SP140	SP140_HUMAN	98,224/5.2	SP140	35.60%	1.39 × 10^9^	defense response
1166	O14730	Serine/threonine-protein kinase RIO3	RIOK3_HUMAN	59,094/5.5	RIOK3	26.40%	4.31 × 10^6^	positive regulation of interferon-beta production, cellular response to dsDNA, innate immune response
1553	Q13563	Polycystin-2	PKD2_HUMAN	109,692/5.5	PKD2	19.90%	1.20 × 10^8^	cellular response to osmotic stress, regulation of cell proliferation, positive regulation of cell cycle arrest
1421	O43182	Rho GTPase-activating protein 6	RHG06_HUMAN	105,948/7.0	ARHGAP6	27.60%	9.16 × 10^7^	inflammatory response, transmembrane transport
Proteins up-regulated by NnV								
2250	O75330	Hyaluronan mediated motility receptor	ZN33B_HUMAN	84,101/5.7	HMMR	35.5%	5.68 × 10^9^	Hyaluronan catabolic process, regulation of G2/M transition of mitotic cell cycle
1562	P15144	Aminopeptidase N	AMPN_HUMAN	109,541/5.3	ANPEP	34.20%	5.11 × 10^11^	cell differentiation, proteolysis
1978	Q86VD1	MORC family CW-type zinc finger protein 1	MORC1_HUMAN	112,882/8.1	MORC1	20.80%	8.49 × 10^6^	negative regulation of transposition, cell differentiation, multicellular organism development
2020	P51784	Ubiquitin carboxyl-terminal hydrolase 11	UBP11_HUMAN	109,818/5.3	USP11	23.90%	1.19 × 10^10^	ubiquitin-dependent protein catabolic process, protein deubiquitination
1378	O00206	Toll-like receptor 4	TLR4_HUMAN	95,681/5.9	TLR4	31.90%	6.08 × 10^8^	activation of MAPK activity, B cell proliferation involved in immune response
2080	Q8N392	Rho GTPase-activating protein 18	RHG18_HUMAN	96,255/7.3	ARHGAP18	21.40%	8.53 × 10^6^	actin filament organization, phagocytosis, engulfment
2514	Q86UV5	Ubiquitin carboxyl-terminal hydrolase 48	UBP48_HUMAN	119,033/5.7	USP48	21.50%	8.78 × 10^7^	protein deubiquitination, ubiquitin-dependent protein catabolic process
1696	Q969V6	MKL/myocardin-like protein 1	MKL1_HUMAN	98,920/5.6	Q969V6	28.20%	1.57 × 10^7^	actin cytoskeleton organization, smooth muscle cell differentiation
1929	Q9Y4L1	Hypoxia up-regulated protein 1	HYOU1_HUMAN	111,336/5.2	Q9Y4L1	31.20%	1.45 × 10^10^	cellular response to Hypoxia, response to ischemia, receptor-mediated endocytosis
1590	Q14587	Heat shock protein 105 kDa	ZN268_HUMAN	96,866/5.3	ZNF268	23.00%	5.01 × 10^7^	receptor-mediated endocytosis, regulation of cellular response to heat, positive regulation of NK T cell activation
1198	Q96FS4	Signal-induced proliferation-associated protein 1	SIPA1_HUMAN	112,150/6.2	SIPA1	15.90%	1.91 × 10^6^	cell proliferation, negative regulation of cell growth, negative regulation of cell cycle
1555	Q5T7N2	LINE-1 type transposase domain-containing protein 1	LITD1_HUMAN	98,850/4.9	L1TD1	18.20%	1.50 × 10^6^	transposition, RNA-mediated
1886	P15918	V(D)J recombination-activating protein 1	RAG1_HUMAN	119,098/8.9	RAG1	15.8%	1.42 × 10^6^	B cell differentiation, T cell homeostasis, negative regulation of thymocyte apoptotic process
710	A6NI28	Rho GTPase-activating protein 42	RHG42_HUMAN	98,570/8.2	ARHGAP42	26.9%	2.94 × 10^7^	activation of GTPase activity, negative regulation of vascular smooth muscle contraction
2199	P21709	Ephrin type-A receptor 1	EPHA1_HUMAN	108,128/6.2	EPHA1	22.0%	3.45 × 10^9^	negative regulation of cell migration, cell surface receptor signaling pathway
1389	Q93033	Immunoglobulin superfamily member 2	IGSF2_HUMAN	115,109/6.5	CD101	22.9%	1.59 × 10^12^	cell surface receptor signaling pathway, positive regulation of myeloid leukocyte differentiation
2305	Q4V348	Zinc finger protein 658B	Z658B_HUMAN	94,332/8.9	ZNF658B	25.40%	1.51 × 10^7^	bile acid biosynthetic process
1122	Q02156	Protein kinase C epsilon type	KPCE_HUMAN	83,675/6.7	PRKCE	30.10%	1.18 × 10^8^	apoptotic process, apoptotic process,
1491	P49736	DNA replication licensing factor MCM2	MCM2_HUMAN	101,897/5.3	MCM2	26.0%	9.15 × 10^8^	apoptotic process, apoptotic process, DNA replication initiation, nucleosome assembly
1827	P35527	Keratin, type I cytoskeletal 9	K1C9_HUMAN	62,065/5.1	KRT9	20.50%	1.04 × 10^6^	intermediate filament organization, cornification

^1^ Accession numbers predicted by Swiss-Prot. ^2^ Theoretical mass (MW) and Pi reported in Swiss-Prot. ^3^ Percentage of amino acids sequence coverage of matched peptides for the identified proteins.

## Data Availability

Not applicable.

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
