# Peer review of "Proteomic Changes during the Dermal Toxicity Induced by Nemopilema nomurai Jellyfish Venom in HaCaT Human Keratinocyte"

_toxins, 2021, doi:10.3390/toxins13050311_

Round 1

Reviewer 1 Report

This manuscript is interesting and possess scientific value. I hope to utilize the results for treatment.

Author Response

Thanks for your comments. We have improved the quality of manuscript after the review comments.

Reviewer 2 Report

Looks an interesting study but the English needs looking at. There are some sentences that do not read correctly. I have some minor comments below:

No mention of the genome paper in the introduction or discussion:

Kim, HM., Weber, J.A., Lee, N. et al. The genome of the giant Nomura’s jellyfish sheds light on the early evolution of active predation. BMC Biol 17, 28 (2019). https://doi.org/10.1186/s12915-019-0643-7

Do the authors have any idea what the dose is of the venom given to a person when they are stung by this jellyfish? How does this relate to the doses used in the experiment?

No magnification or scale bars on figures 2 and 3 or mentioned in the methods.  

No masses or pI’s labelled on figure 4 (see Choudhary et al 2019 from toxins to show how it should be done)

Was it quantitative silver staining used to stain the 2D gels I do see that it is mass spectrometry compatible?

Table 1 doesn’t seem to have any headings and maybe should be an appendix rather than in the main document, or set out as in Choudhary et al 2019? The numbers underneath the paper don’t mean anything as there are no numbers in the table, have the headings been cut off? (see Choudhary et al 2019 from toxins to show how it should be done)

Author Response

< Reviewer #2 >

1)  No mention of the genome paper in the introduction or discussion:

Reply: Thanks for your comments, now we have provided genome paper reference in introduction section.

2)  Do the authors have any idea what the dose is of the venom given to a person when they are stung by this jellyfish? How does this relate to the doses used in the experiment?

Reply: The question is propably one of the key issues in understanding and predicting the toxinological effects of poisonous jellyfish venoms. Unfortunately, however, we do not have a clear answer for it, since there can be so many variables affecting the toxicosis, including how much large the surface area of stinged region, how long the sting happened, what kinds of poisonous jellyfish species associated, and so on. For the present study, we have used in vitro cell culture model, hence the actual dose of venom sting had not been counted on.

3)  No magnification or scale bars on figures 2 and 3 or mentioned in the methods.

Reply: Thanks for your comments. We have provided scale bars on figures 2 and 3

4)  No masses or pI’s labelled on figure 4 (see Choudhary et al 2019 from toxins to show how it should be done)

Reply: We agree with the comments. We have changed accordingly.

5)            Was it quantitative silver staining used to stain the 2D gels I do see that it is mass spectrometry compatible?

Reply: Thanks for your comments. Quantitative silver staining was used to stain the 2D gels. We have not seen the absolute amount of each protein spot, but instead we have observed the relative change of the spot upon the treatment. For this, three biological replicates of the experiment were performed to obtain the reproducibility of the gels. All 2-DE gels were further scanned and analyzed using Progenesis Same Spots software.

6)   Table 1 doesn’t seem to have any headings and maybe should be an appendix rather than in the main document, or set out as in Choudhary et al 2019? The numbers underneath the paper don’t mean anything as there are no numbers in the table, have the headings been cut off? (see Choudhary et al 2019 from toxins to show how it should be done)

Reply: We agree with the comments. We have changed accordingly

Reviewer 3 Report

Comments and Suggestions for Authors

toxins-1165337-peer-review-v1

The article entitled “Proteomic changes during the dermal toxicity induced by Nemopilema nomurai jellyfish venom in HaCaT human keratinocyte” tackles a very interesting point related with the changes produced by N. nomurai stings upon skins cells and the identification of putative therapeutic targets. For this purpose, the author extracted the cnematocysts extract (NvN), and different protein concentration (0-5 ug/mL) were used to assess the cytotoxic effects of the venom in HaCaT keratinocyte Cell Lines. Then, the authors determined the IC50 value (2.5 μg/mL) of NnV that was used for treating HaCaT cells for proteomic analysis. The MALDI-TOF/MS analyses revealed a total of 44 differentially expressed proteins (DEPs, or differentially abundant proteins (DAPs)) from 70 protein spots with significant quantitative changes obtained from the 2DGE, of which 19 proteins DEPs/DAPs showed an increased expression level, whereas 25 proteins (DEPs/DAPs?) decreased compared to the control. Some of these DEPs/DAPs were identified as Plasminogen, Vinculin, EMILIN-1, Basonuclin2, Focal adhesion kinase 1, FAM83B, Peroxisome proliferator-activated receptor-gamma co-activator 1-alpha, stress or immune response-related proteins like Toll-like receptor 4, Aminopeptidase N, MKL/myocardin-like protein 23 1, Hypoxia up-regulated protein 1, Heat shock protein 105 kDa, Ephrin type-A receptor 1. The DEPs/DAPs were then functionally annotated according to Gene ontology (GO) categories in four group (molecular function, biological process, cellular component and protein class) using the Phanther classification system (http:www.pantherdb.org/), and their putative interaction network were also explored using STRING database V 10.5 (http://string-db.org).

Comments/Questions:

Abstract:

Line 17: The authors used the terms differentially expressed proteins (DEPs) to the proteins that showed significant changes in expression/volume/abundance, and sometimes in the text used the terms differentially abundant proteins (DAPs). I would like the authors to use the same nomenclature for these proteins. Perhaps, DPAs is the correct one, since is more related with volume/spots/2DE.

Lines 19-22 Why the authors classify the downregulated proteins as required for cell survival and development (i.e. Plasminogen, Vinculin, EMILIN-1, Basonuclin2, Focal adhesion kinase 1, FAM83B, Peroxisome proliferator-activated receptor-gamma co-activator 1-alpha). What about the other proteins, are not essential for cell survival? Please, I would like the authors to explain more about this classification?

Besides, would be better to check if the previous idea is correct written, since at the end of the sentences (line 19: he down-regulated proteins were, and in the same sentence line 22: were decreased in abundance after NnV treatment), seems to be redundant.

The last sentences of the abstract can be improved as well, giving information about the conclusions.

Introduction

The Introduction chapter provides a proper state-of-the art of the dermal toxicity of venom throughout the animal kingdom. However, less information about dermal toxicity of jellyfishes’ sting is provided, compared to other venomous animals.

I also would like to recommend the authors to move some information for the introduction to discussion, or just use it in the discussion section as needed. In this respect, my recommendation is to limit some of the information provided from line 58 to line 79)

Just provide more information about similar experimental model with yours, previously used to study the dermal toxicity of jellyfishes (or cnidarians s.l.) or animal venoms.

Please follow below my suggestion in a more specific way:

Dermal toxicity is prevalent in response to venom from different organisms, and the mode of action is almost the same, producing pain, hemorrhagic activity, bleeding, mast cell degranulation, edema, disruption of cell metabolism, inflammatory response, and necrosis (refs). Among the compounds involved in the injured skin effects have been reported metalloproteinase, phospholipase D, hyaluronidase, phosphatases, histamine, serotonin, hemolysins... (refs).  Our recent proteomics studies revealed that Nemopilema nomurai venom contains various novel components such as phospho- lipase D Li Sic Tox beta IDI, a serine protease, putative Kunitz-type serine protease inhib- itor, phospholipase A2, disintegrin and metalloproteinase, leukotoxin, hemolysin, three- finger toxin MALT0044C, allergens, venom prothrombin activator trocarin D, tripeptide Gsp 9.1, and many other toxin proteins [22]. Such components may contribute to venom toxicity and leads to the deleterious effect of venom on human health.

Line 61: Latin names must be italicized (see Loxosceles)

Results

Results are well presented giving a good description of the main findings, but the quality of Figures, Tables, and supplementary must be checked before publication.

Lines 117-118 (Figure 1): Please check the quality of the Figure 1, since there are some blurred numbers likely related with the concentration applied.

Line 161 (Table 1): The information of the header is missing, confused in the Table 1. The format of the Table 1 must be checked, and please consider avoiding colors (grey, black) in the cells background (see also Table 2 format).

The authors provided the “Percentage of amino acids sequence coverage of matched peptides for the identified proteins”. I would like to know how many peptides (number of peptides) per proteins were identified, and their corresponding sequences. This information should be included in Table 1.

Discussion:

In my opinion the discussion needs to be improved.

The author should try to comment/discuss the mechanism of action underlying the venom toxicity, and/or to interconnect the role of the DEPs/DAPs to the toxic effects observed in this model (HaCaT cells lines) in a different way than a lineal description. The authors give some information/description of the biological function/activity of each DEP/DAPs found, but less focused on how the presence of these proteins can help to explain the toxic effects observed/reported. Besides, each DEP/DAP description ended with the idea/conclusion that XXX component leads/cause toxicity/severe damage in HaCaT cells. Thus, this particular way of discussion needs to be improved/enriched.

I would like the authors to use more the Table 2 to explore the networks, eventually synergically effects produced by the venom components in the model studied. Eventually, the such mechanism/hypothesis discussed can be extrapolated to the real-life involving jellyfish sting on human.

Line 217: this idea needs to be improved “MALDI/TOF/MS was used to identify the proteins”. What proteins, venom DEPs? I understand the authors referred to DEPs, but the idea must be improved anyway to be understandable for readers.  

Conclusions:

I think the conclusion must be improved highlighting the main finding in line with the aims. Therefore, I´m not sure the author wrote the conclusion properly, wrote “In conclusion, the present findings may exhibit some possible key players during the skin damages and suggest therapeutic strategies for the prevention of jellyfish envenomation”. However, there is no clear conclusions about the mechanism of envenomation, nor a convincing conclusion that allow the authors to address any therapeutic strategy. I would like the authors to explain those therapeutic strategies clearly, or just improved the conclusion/remarks.

Materials and Methods:

The Material and Methods are clearly presented, providing a detailed description of the methodology applied.

My only concern is about how the authors determined the presence of cnidarians toxins and/or venom extract composition?

The authors described in Line 406:  Bradford assay was explored (Bio-Rad, CA, USA) [62] to determine the protein concentration of venom samples, but no more information whether the venom extract was analysed by MALDI- TOF /MS analyses.

Considering the putative degradation of some venom components, and the variations previously reported in Nemopilema nomurai (e.g. seasonal, geographical), I think the readers would like to know the venom composition of the extract studied, or at least more clearly discussed if the composition were previously studied/reported (I suppose it was).

Author Response

< Reviewer #3 >

1)  Line 17: The authors used the terms differentially expressed proteins (DEPs) to the proteins that showed significant changes in expression/volume/abundance, and sometimes in the text used the terms differentially abundant proteins (DAPs). I would like the authors to use the same nomenclature for these proteins. Perhaps, DAPs is the correct one, since it is more related with volume/spots/2DE.

Reply: Thanks for your comments. We have changed accordingly throughout the revised manuscript.

2)            Lines 19-22 Why the authors classify the downregulated proteins as required for cell survival and development (i.e. Plasminogen, Vinculin, EMILIN-1, Basonuclin2, Focal adhesion kinase 1, FAM83B, Peroxisome proliferator-activated receptor-gamma co-activator 1-alpha). What about the other proteins, are not essential for cell survival? Please, I would like the authors to explain more about this classification?

Reply: Our data show that the proteins which are required for the cell survival and development were downregulated, whereas the stress marker proteins were upregulated after NnV treatment. Although the whole spectrum of cellular events and their cause-effect relation upon NnV treatment are yet to be determined, our results clearly support the predictable toxic effects of NnV on HaCaT cells. We have addressed this issue in Results and Discussion parts.

3)   Besides, would be better to check if the previous idea is correct written, since at the end of the sentences (line 19: he down-regulated proteins were, and in the same sentence line 22: were decreased in abundance after NnV treatment), seems to be redundant.

Reply: Thanks for your comments. We have changed accordingly.

4)  The last sentences of the abstract can be improved as well, giving information about the conclusions.

Reply: Thanks for your comments. We have changed accordingly.

5)  The Introduction chapter provides a proper state-of-the art of the dermal toxicity of venom throughout the animal kingdom. However, less information about dermal toxicity of jellyfishes’ sting is provided, compared to other venomous animals.

Reply: Thanks for your comments. We have changed the introduction as per suggestion.

6)  I also would like to recommend the authors to move some information for the introduction to discussion, or just use it in the discussion section as needed. In this respect, my recommendation is to limit some of the information provided from line 58 to line 79).

Reply: Thanks for your comments. I agree with you and I have changed the introduction as per suggestion

7)  Just provide more information about similar experimental model with yours, previously used to study the dermal toxicity of jellyfishes (or cnidarians s.l.) or animal venoms.

Reply: Thanks for your comments. We agree with you and I have provided our previous dermal toxicity in vivo studies and other jelly fish dermal toxicity in the introduction.

8)  Please follow below my suggestion in a more specific way:

Dermal toxicity is prevalent in response to venom from different organisms, and the mode of action is almost the same, producing pain, hemorrhagic activity, bleeding, mast cell degranulation, edema, disruption of cell metabolism, inflammatory response, and necrosis (refs). Among the compounds involved in the injured skin effects have been reported metalloproteinase, phospholipase D, hyaluronidase, phosphatases, histamine, serotonin, hemolysins... (refs).  Our recent proteomics studies revealed that Nemopilema nomurai venom contains various novel components such as phospho- lipase D Li Sic Tox beta IDI, a serine protease, putative Kunitz-type serine protease inhib- itor, phospholipase A2, disintegrin and metalloproteinase, leukotoxin, hemolysin, three- finger toxin MALT0044C, allergens, venom prothrombin activator trocarin D, tripeptide Gsp 9.1, and many other toxin proteins [22]. Such components may contribute to venom toxicity and leads to the deleterious effect of venom on human health.

Reply: Thanks for your comments. I agree with you and I provided the references for the following text.

9)  Line 61: Latin names must be italicized (see Loxosceles).

Reply: Thanks for your comments. We have changed accordingly.

10)    Results are well presented giving a good description of the main findings, but the quality of Figures, Tables, and supplementary must be checked before publication.

Reply: Thanks for your comments. We have changed accordingly.

11)   Lines 117-118 (Figure 1): Please check the quality of the Figure 1, since there are some blurred numbers likely related with the concentration applied.

Reply: Thanks for your comments. We have changed accordingly.

12)     Line 161 (Table 1): The information of the header is missing, confused in the Table 1. The format of the Table 1 must be checked, and please consider avoiding colors (grey, black) in the cells background (see also Table 2 format).

Reply: Thanks for your comments. We have changed accordingly.

13)     The authors provided the “Percentage of amino acids sequence coverage of matched peptides for the identified proteins”. I would like to know how many peptides (number of peptides) per proteins were identified, and their corresponding sequences. This information should be included in Table 1.

Reply: Thanks for your comments. These experiments were done 3 years earlier and that time we have used Protein prospector software to analyze MALDI/TOF/MS data. At that time we followed previously published papers to extract the data in which peptide sequence and number of peptides were not shown. Also it is not possible to provide the sequences of each peptide for DEPs. For your reference we have added one examples below.

14)     In my opinion the discussion needs to be improved. The author should try to comment/discuss the mechanism of action underlying the venom toxicity, and/or to interconnect the role of the DEPs/DAPs to the toxic effects observed in this model (HaCaT cells lines) in a different way than a lineal description. The authors give some information/description of the biological function/activity of each DEP/DAPs found, but less focused on how the presence of these proteins can help to explain the toxic effects observed/reported. Besides, each DEP/DAP description ended with the idea/conclusion that XXX component leads/cause toxicity/severe damage in HaCaT cells. Thus, this particular way of discussion needs to be improved/enriched.

Reply: Thanks for your comments. We have improved the introduction section and conclusion

15)     I would like the authors to use more the Table 2 to explore the networks, eventually synergically effects produced by the venom components in the model studied. Eventually, the such mechanism/hypothesis discussed can be extrapolated to the real-life involving jellyfish sting on human.

Reply: We have followed the above suggestions and incorporated in the text.

16)     Line 217: this idea needs to be improved “MALDI/TOF/MS was used to identify the proteins”. What proteins, venom DEPs? I understand the authors referred to DEPs, but the idea must be improved anyway to be understandable for readers.

Reply: Thanks for your comments. We have changed accordingly.

17)     I think the conclusion must be improved highlighting the main finding in line….

Reply: Thanks for your valuable comments and suggestion. We have improved the discussion and conclusion section of the manuscript highlighted

18)     The authors described in Line 406: Bradford assay was explored (Bio-Rad, CA, USA) [62] to determine the protein concentration of venom samples, but no more information whether the venom extract was analysed by MALDI- TOF /MS analyses.

Reply: Thank you very much for the precious comments and suggestions. In our previous study, we have already characterized the Nemopilema nomurai venom components. Altogether, 150 proteins were identified, comprising toxins and other distinct proteins that are substantial in nematocyst genesis and nematocyte growth by employing two-dimensional gel electrophoresis and matrix-assisted laser desorption/ionization time of flight mass spectrometry (MALDI/TOF/MS) https://doi.org/10.3390/toxins11030153.

19)     Considering the putative degradation of some venom components, and the variations previously reported in Nemopilema nomurai (e.g. seasonal, geographical),  I think the readers would like to know the venom composition of the extract studied, or at least more clearly discussed if the composition were previously studied/reported (I suppose it was).

Reply: Thanks for your comments. We have previously studied NnV components by using proteomics approach (https://doi.org/10.3390/toxins11030153), which has been briefly addressed and cited in the present manuscript.

Round 2

Round 2

Dear authors,

Thanks for the improvement made to the manuscript.

However, the manuscript still needs to be polished.

Minor comments:

Line 19: Perhaps, this sentence is too long. Please, check it again and edit it if possible. Consider this suggestion:

DAPs involved in cell survival and development (e.g., Plasminogen, Vinculin, EMILIN-1, Basonuclin2, Focal adhesion kinase 1, FAM83B, Peroxisome proliferator-activated receptor-gamma co-activatorcoactivator 1-alpha) decreased their expression, whereas stress or immune response-related proteins (e.g., Toll- like receptor 4, Aminopeptidase N, MKL/myocardin-like protein 1, Hypoxia up-regulated protein 1, Heat shock protein 105 kDa, Ephrin type-A receptor 1, with some protease (or peptidase) enzymes) were up-regulated.

Introduction

....

From line 51 to line 93

I don’t think this text help the readers to understand the history, instead distort the main idea that led the aims. So, the author can remove the text below, or use it in the discussion if needed.

Moreover, histamine is released due to the exocytosis of mast cells by jellyfish venom [89]. Jellyfish stings can also result in many systemic effects symptoms such as neurological gastroin- testinal, cardiac, or allergic responses after entering the general circulation [910]. Ventric- ular arrhythmias and cardiac arrest due to jellyfish venom in severe cases may lead to mortalities [10-1211-13]. Acute renal failure was also observed due to intravascular he- molysis by jellyfish envenomation [1112].. Our former investigation has shown that the metalloproteinase present in the scyphozoan jellyfish venom predominately mediates dermal toxicity [14]. Previous animal model studies also revealed that metalloproteinase plays a crucial role in damaged skin tissue's clinical manifestations [14,15]. The box jelly- fish Chironex fleckeri is the most venomous marine creature, and its envenomation can cause cardiorespiratory distress and leads to death within few minutes [16,17]. Jellyfish Chironex fleckeri toxins can induce edema, vesicle formation, erythema, result in extensive progression of necrosis and cause purple to brown wounds[16,17,18]. The everlasting se- rious impediment of such wounds consists of granulomas, hyperpigmentation, fat atro- phy, and keloids[17,18]. NnV is the fusion of complex components abundant in peptides and proteins, which are discharged after proper provocation[19].

.... LTX exposure resulted in severe dermatological distress during Ostreopsis blooms; hence PLTX triggers the cytotoxic effects in vitro [2125].

If the authors agree, can follow this suggestion to end the introduction:

Our former investigation has shown that the metalloproteinase present in the scyphozoan jellyfish venom predominately mediates dermal toxicity [14]. However, the mechanism of dermal toxicity involving NnV effects remain unexplored. NnV is the fusion of complex components abundant in peptides and proteins, which are discharged after proper provocation [19]. Our recent proteomics studies revealed that Nemopilema nomurai venom contains various novel components such as phospholipase D Li Sic Tox beta IDI, a serine protease, putative Kunitz-type serine pro- tease inhibitor, phospholipase A2, disintegrin and metalloproteinase, leukotoxin, hemo- lysin, three-finger toxin MALT0044C, allergens, venom prothrombin activator trocarin D, ....

Line 414: Provide the information of this final paragraph in a separated section as Conclusions or maybe Remarks.

Author Response

<Reviewer #3>

1] Line 19: Perhaps, this sentence is too long. Please, check it again and edit it if possible. Consider this suggestion:

à Reply: Thanks for your comments. We have changed accordingly.

2] Line 51 to line 93: I don’t think this text help the readers to understand the history, instead distort the main idea that led the aims. So, the author can remove the text below, or use it in the discussion if needed.

à Reply: Thanks for your comments. We have changed some part of introduction according to reviewer comments.

If the authors agree, can follow this suggestion to end the introduction:

Our former investigation has shown that the metalloproteinase present in the scyphozoan jellyfish venom predominately mediates dermal toxicity [14]. However, the mechanism of dermal toxicity involving NnV effects remain unexplored. NnV is the fusion of complex components abundant in peptides and proteins, which are discharged after proper provocation [19]. Our recent proteomics studies revealed that Nemopilema nomurai venom contains various novel components such as phospholipase D Li Sic Tox beta IDI, a serine protease, putative Kunitz-type serine pro- tease inhibitor, phospholipase A2, disintegrin and metalloproteinase, leukotoxin, hemo- lysin, three-finger toxin MALT0044C, allergens, venom prothrombin activator trocarin D, ....

à Reply: Thanks for your great comments and suggestions. We have changed introduction accordingly.

3] Line 414: Provide the information of this final paragraph in a separated section as Conclusions or maybe Remarks.

à Reply: Thanks for your comments. We have changed accordingly.